# Comprehensive Analysis of Oncogenic Somatic Alterations of Mismatch Repair Gene in Breast Cancer Patients

**DOI:** 10.3390/bioengineering12040426

**Published:** 2025-04-18

**Authors:** Yin Yan, Yang Wang, Junjie Tang, Xiaoran Liu, Jichuan Wang, Guohong Song, Huiping Li

**Affiliations:** 1Key Laboratory of Carcinogenesis and Translational Research (Ministry of Education), Department of Breast Oncology, Peking University Cancer Hospital & Institute, Beijing 100142, China; yanying_pku@163.com (Y.Y.); liuxiaoran@bjmu.edu.cn (X.L.); 2Key Laboratory of Carcinogenesis and Translational Research (Ministry of Education), Comprehensive Clinical Trial Ward, Peking University Cancer Hospital & Institute, Beijing 100142, China; wangyang@bjmu.edu.cn; 3The First Clinical Medical School, Nanjing Medical University, Nanjing 211166, China; tjj_njmu@163.com; 4Musculoskleletal Tumor Center, Beijing Key Laboratory for Musculoskeletal Tumors, Peking University People’s Hospital, Beijing 100041, China; jcwang@pku.edu.cn

**Keywords:** mismatch repair genes, breast cancer, somatic alterations, tumor mutational burden, intratumoral microbiome

## Abstract

Recent clinical trials have suggested that solid cancers with mismatch repair (MMR) deficiency are highly responsive to immunotherapy, regardless of cancer types. Previous MMR-related studies on breast cancer have predominantly focused on germline variants. However, the somatic MMR alterations have not been comprehensively characterized in breast cancer. In this study, we integrated genomic, transcriptomic, and clinical data from over 3000 breast cancer cases across six public cohorts. Our findings revealed that 1.2% of breast cancers harbored oncogenic somatic MMR alterations, with triple-negative breast cancer (TNBC) demonstrating the highest mutation rate at 3.1%. Additionally, somatic MMR alterations were significantly associated with microsatellite instability-high (MSI-H) and MMR-related mutational signatures, indicating that somatic MMR alterations led to impaired function of the MMR system. Biallelic inactivation of MMR genes resulted in a more pronounced loss of MMR function compared to monoallelic inactivation. Importantly, these MMR alterations significantly increased the tumor mutational burden (TMB) and neoantigen load in breast cancer, regardless of MSI-H status. These findings indicate that the frequency of MMR alterations is highest in TNBC and that MMR alterations in breast cancer can lead to MMR functional deficiencies, suggesting that some patients harboring such alterations may benefit from immunotherapy.

## 1. Introduction

Mismatch repair (MMR) genes are critical susceptibility genes for Lynch syndrome, with deficiencies linked to an increased risk of colorectal, endometrial and gastric cancers, and other malignancies [1]. MMR gene deficiencies in Lynch syndrome-associated tumors affect clinicopathological features, including poor differentiation, high lymphocytic infiltration, and favorable prognosis [2]. Additionally, MMR deficiency in these tumors leads to the loss of MMR proteins (dMMR), accumulation of somatic mutations, especially high instability in microsatellite sites, and activating the tumor immune microenvironment [3], which renders these tumors sensitive to immunotherapy [4].

Clinical trials have highlighted the significant response of MMR-deficient solid tumors to immunotherapy, regardless of tumor type [5], which has aroused increased interest in exploring MMR deficiencies in various cancers. Breast cancer is not classically part of the Lynch syndrome tumor spectrum [6], and studies on MMR gene deficiencies in breast cancer remain limited, with most focusing on germline variants [7,8,9,10,11,12,13]. These studies have shown that germline variants in MMR genes are extremely rare in unselected breast cancer [12,14]. Nonetheless, these germline MMR variants were associated with MMR protein loss, microsatellite instability-high (MSI-H), elevated tumor mutational burden (TMB-H), and increased PD-L1 expression [10,11,12,13,15]. Moreover, breast cancers harboring MMR germline variants exhibit distinct clinical characteristics compared to their wildtype counterparts, including a higher prevalence of high tumor grade and worse prognosis [10,11,12,15,16]. However, somatic alterations in MMR genes, such as somatic mutations, copy number variations, or structural variations, have not been comprehensively characterized in breast cancer.

To address this gap, we integrated genomic, transcriptomic, and clinical data from over 3000 breast cancer cases across six public cohorts to identify oncogenic somatic alterations in MMR genes. Our findings identified somatic MMR alterations in 1.2% of breast cancers, with the highest prevalence observed in triple-negative breast cancer (TNBC) (3.1%). These alterations were significantly associated with MSI-H status and MMR-related mutational signatures. Notably, somatic MMR alterations were linked to a significant increase in TMB, independent of MSI-H status. Moreover, MMR deficiencies in breast cancer may reshape the tumor mutational landscape and modulate the composition of the intratumoral microbiome.

## 2. Methods

### 2.1. Online Data Acquisition

In this study, we integrated clinical data and next-generation sequencing (NGS) data from six public breast cancer cohorts [17,18,19,20,21,22]. After removing duplicate cases, a total of 3667 breast cancer patients were included in the analysis (Appendix A). Tumor samples from all patients across the six cohorts underwent panel sequencing (including MLH1, MSH2, MSH6, PMS2 genes) or whole-exome sequencing (WES). Clinical characteristics and NGS data were obtained from the cBioPortal database (https://www.cbioportal.org, accessed on 1 August 2024) or the original publications.

### 2.2. Identification of Oncogenic Somatic Alterations in MMR Genes

In this study, we reanalyzed the somatic mutations in MMR genes detected across the six breast cancer cohorts. First, high-quality somatic mutations in MMR genes that met the following criteria were selected: sequencing depth greater than 20× and variant allele frequency (VAF) exceeding 10%. We further filtered mutations based on the following criteria: (1) truncating variants (nonsense and frameshift mutations) were included, but those located within the last 55 base pairs of the penultimate or final exon, which may escape nonsense-mediated mRNA decay and do not impact known functional domains, were excluded; (2) essential splice variants or missense mutations that were identified as loss-of-function in published literature or annotated as oncogenic by the OncoKB tool [23]. In addition, somatic homozygous deletions and structural variants (SV) in MMR genes were included for further analysis, as these alterations can also result in loss of function of MMR genes. For simplicity, we refer to oncogenic somatic mutations, homozygous deletions, and SV in MMR genes collectively as “MMR-altered” in this study.

### 2.3. Transcriptomic Data Analysis

Bulk RNA sequencing (RNA-seq) was performed on tumor tissues from 1176 patients in two of the cohorts [17,22]. We filtered and merged mRNA expression data based on common genes across cases. These data were batch-corrected using the limma package, standardized using the log(x − min(x) + 1) method, and then subjected to downstream analysis.

### 2.4. Gene Set Enrichment Analysis (GSEA)

GSEA [24] was performed on hallmark pathways (v7.0) and the KEGG dataset to identify the biological functional differences between MMR-altered and MMR-wt breast cancers. The result was presented using the “ggplot2” (v 3.3.3) R package. *p* value < 0.05 was considered to be statistically significant.

### 2.5. Inference of Infiltrating Immune Cells

To evaluate the relationship between MMR-altered and infiltrating immune cells in breast cancers, normalized gene expression data were used to infer the relative proportions of 22 types of infiltrating immune cells using the CIBERSORT algorithm. The 22 cell types inferred by CIBERSORT encompass B cells naive, B cells memory, plasma cells, T cells CD8, T cells CD4 naive, T cells CD4 memory resting, T cells CD4 memory activated, T cells follicular helper, T cells regulatory (Tregs), T cells gamma delta, NK cells resting, NK cells activated, monocytes, macrophages M0, macrophages M1, macrophages M2, dendritic cells resting, dendritic cells activated, mast cells resting, mast cells activated, eosinophils, and neutrophils.

### 2.6. Immunohistochemistry (IHC) Assay of MMR Protein

IHC was performed on 4 μm sections of formalin-fixed, paraffin-embedded tissue from a primary breast cancer case harboring a pathogenic somatic mutation in the MLH1 gene (p.E102*). This case was derived from Breast Oncology Department of Peking University Cancer Hospital, and the MLH1 mutation was identified through clinical genetic testing. Informed written consent was obtained from this participant. This study was performed in accordance with the Declaration of Helsinki and was approved by the Research and Ethics Committee of Peking University Cancer Hospital (2020KT75). Detection of MMR proteins was conducted using primary antibodies against MLH1 (clone GM002, a mouse monoclonal antibody, Gene Tech Co., Ltd. Shanghai, China), MSH2 (clone RED2, rabbit monoclonal antibody, Gene Tech Co., Ltd. Shanghai, China), MSH6 (clone EP49, rabbit monoclonal antibody, Gene Tech Co., Ltd. Shanghai, China), and PMS2 (clone EP51, rabbit monoclonal antibody, Gene Tech Co., Ltd. Shanghai, China). Loss of MMR protein expression was defined as the absence of nuclear immunostaining for one or more MMR proteins in tumor cells on whole section slides.

### 2.7. Mutational Signatures

Samples with at least 50 somatic single nucleotide variants (SNVs) were subjected to mutational signature analysis. Mutational signatures were inferred from non-synonymous and silent somatic exonic SNVs using the deconstructSigs algorithm [25], based on the 30 mutational signatures cataloged in COSMIC [26]. The dominant mutational signature for each sample was defined as any signature contributing more than 10% of the total mutational signature.

### 2.8. Biallelic Inactivation Analysis

To assess biallelic inactivation events, both somatic mutations and copy number alterations (CNAs) were analyzed. For somatic mutations, the VAF was first calculated as the ratio of the tumor alternate allele count (t_alt_count) to the total tumor coverage (t_ref_count + t_alt_count), based on genomic sequencing data obtained from public databases. The raw VAF values were then adjusted for tumor purity (TP) by dividing VAF by TP. Loss of heterozygosity (LOH) was defined as a corrected VAF/TP value ≥ 0.9, the commonly used cut-off for LOH [27]. For CNAs, biallelic inactivation was identified when the CNA value < −1, indicative of a deep deletion event likely to affect both alleles.

### 2.9. TMB, Neoantigens, MSI and MMRDetect Analysis

TMB is defined as the total number of nonsynonymous somatic mutations, encompassing SNVs and small insertions/deletions, per mega base of coding regions. The TMB for each tumor sample is reported as mutations per mega base (Mut/Mb). A TMB threshold of ≥ 10 Mut/Mb is used to classify tumors as TMB-H. Neoantigens are tumor-specific peptides generated by somatic mutations, the load of which is often correlated with TMB. Previous studies have systematically predicted the neoantigen load from The Cancer Genome Atlas (TCGA) data by integrating tools such as OptiType, NetMHCpan, and pVAC-Seq [28]. Peptides predicted to bind with MHC proteins (pMHCs) were identified from SNVs and indel mutations. The number of pMHCs per megabase (pMHCs/Mb) was then used as a measure of the neoantigen load. For WES data from the TCGA cohort, the calculation of MSI followed the methodologies outlined in the literature [29]. The MSI status for the MSK-IMPACT sequencing data was derived from the original publication [20]. Additionally, a study developed a logistic regression-based classifier for MMR deficiency, named MMRDetect, based on mutational signatures derived from experimental data [30]. Since this classifier requires WES/whole-genome sequencing (WGS) data, we applied it to systematically analyze the mutation data from TCGA. Finally, MMR deficiency was defined based on the classifier’s cut-off value of a probability < 0.7, as specified in the original study [30].

### 2.10. Intratumoral Microbiota Analysis

To investigate the intratumoral microbiota in breast cancer, we revisited and retrieved microbial profiles from breast tumors which had been previously processed by Haziza et al. using RNA-Seq data from TCGA [31]. The specific workflow was as follows:

Microbial Read Extraction and Processing: First, RNA reads that did not align to the human reference genome (GRCh38) were reprocessed with additional alignment and quality control. Low-quality reads were removed, and the remaining reads were mapped to the RefSeq database, which includes 11,955 microbial genomes. As a result, microbial feature counts were detected in 15,512 samples. Among the total 6.06 × 10^12^ sequencing reads, 7.3% did not align to the human genome, and 1.2% (0.08% of total reads) were successfully mapped to the microbial database. Of these non-human reads, the majority (80.2%) were classified as bacterial, while the rest included fungi, viruses, and other microbes.

Contamination Removal Strategy: Since TCGA data lacked contamination controls, to ensure that microbial reads originated from tumors rather than environmental contamination, Haziza et al. implemented a two-step decontamination process: (1) In silico contamination removal based on sequencing plate and center: The research team first applied an in silico decontamination method based on sequencing plate and center [32]. Briefly, samples were divided into multiple batches based on their sequencing plate-center combinations. The decontam R package (with *p* * = 0.1) was used to perform linear regression analyses within each batch to assess the likelihood of taxonomic contamination. If a microbial taxon was identified as a contaminant in any batch, it was removed from the dataset. Furthermore, a Bayesian approach was used to simulate pseudo-contaminants and evaluate their contributions to the model, ensuring robust contamination removal. This process ultimately generated a decontaminated dataset for downstream analyses. (2) Cross-referencing with previous human microbiome studies: The team further removed potential contaminants by referencing previous human microbiome studies, such as the WIS decontaminated amplicon sequencing cohort [33]. Specifically, the WIS study implemented a rigorous six-step filtering process to remove contaminants, including eliminating common laboratory contaminants, controlling for DNA extraction, PCR amplification, sequencing batch effects, and potential contamination introduced during paraffin block preparation and storage. The WIS cohort ultimately identified 528 microbial taxa reliably present in tumors. To ensure the reliability of microbial signals in TCGA data, the research team intersected the microbial taxa detected in TCGA with those identified in the WIS cohort, retaining only those taxa detected in both independent datasets. Although this approach was stringent, it maximized the accuracy of microbial data and minimized false positives. Overall, this workflow provided high-quality microbial data for downstream analyses.

Data Availability and Further Processing: Haziza et al. uploaded the fully processed TCGA bacterial and fungal data to their study’s GitHub repository (github.com/knightlab-analyses/mycobiome, accessed on 15 October 2023). This dataset includes microbial read matrices at the genus and species levels (genus-level: 14,494 samples × 250 taxa; species-level: 12,773 samples × 297 taxa), along with corresponding TCGA sample metadata and a taxonomy table. Since the raw data were in raw count format, we applied the ConQuR package [32,34], a recently developed method designed to remove batch effects in microbiome data, to correct for batch effects from sequencing center and experimental strategy (WGS vs. RNA-seq). Finally, we selected breast cancer samples with well-defined MMR gene mutations or MSI status for further analysis (genus-level: 980 samples × 250 taxa).

Quantification of Microbial Presence: The function estimate_richness from the “phyloseq” package was employed to calculate the Shannon index, which reflects the alpha diversity, capturing both the richness and evenness of microbial communities within each sample. Beta diversity, representing the dissimilarity between microbial communities, was assessed using Principal Coordinate Analysis (PCoA) based on Bray–Curtis dissimilarity. To compare differences in microbial communities across groups, permutational multivariate analysis of variance (PERMANOVA) was conducted, also using Bray–Curtis dissimilarity.

### 2.11. Statistical Analysis

Categorical variables were compared using the Chi Square test or the Fisher exact test where appropriate. Continuous variables were tested with a *t*-test or Mann–Whitney U test, where appropriate. Survival was estimated using the Kaplan–Meier method. Log rank test was used to determine whether a factor was associated with survival. Two-sided *p* values less than 0.05 were considered to be statistically significant. All analyses were performed using SPSS 20.0 software.

## 3. Results

### 3.1. Prevalence and Clinical Relevance of Oncogenic Somatic Alterations of MMR Genes in Breast Cancers

A total of 3667 breast cancer patients from six cohorts were included in this study (Figure 1A) (Appendix A). The baseline demographics and clinicopathologic characteristics of these patients are summarized in Table 1. Tumor samples from all patients underwent panel sequencing (including the MLH1, MSH2, MSH6, and PMS2 genes) or WES. Based on the sequencing data, 44 out of 3667 breast cancer cases (1.2%) were found to harbor oncogenic somatic alterations in MMR genes (Figure 1B) (Appendix A). Further analysis of the mutation frequency across different molecular subtypes revealed that MMR gene mutations were more prevalent in TNBC compared to the hormone-receptor-positive/human epidermal growth factor receptor 2-negative (HR+/HER2-) subgroup (3.1% vs. 0.8%, *p* < 0.001, Chi Square test) and the HER2+ subgroup (3.1% vs. 1.3%, *p* = 0.09, Chi Square test) (Table 2). Among these MMR alterations, the majority were somatic mutations (29/44), followed by homozygous copy number depletion (12/44) and SV (3/44) (Figure 1C). The mutation frequencies for the four MMR genes were as follows: MLH1 (11/44), MSH2 (8/44), MSH6 (10/44), and PMS2 (12/44). Additionally, three patients simultaneously harbored somatic mutations in two different MMR genes (Figure 1C).

We further compared the clinicopathologic features and survival outcomes between breast cancers with MMR mutations (hereafter referred to as MMR-altered) and those with wild-type MMR (hereafter referred to as MMR-wt). Compared to MMR-wt breast cancers, MMR-altered breast cancers had a significantly higher proportion of TNBCs (36.4% vs. 13.5%, *p* < 0.001, Chi Square test) and older age at diagnosis (median 57 years vs. 54 years, *p* = 0.01, Mann–Whitney U test) (Table 1, Figure 1D). However, no significant differences were observed between the two groups in terms of ethnicity, histology, TNM stage, or other clinicopathologic features (Table 1). Additionally, there was no significant difference in overall survival (OS) or progression-free survival (PFS) between the two groups (Figure 1E,F).

### 3.2. Somatic Alterations in MMR Genes Result in Impaired MMR Function in Breast Cancers

In Lynch syndrome-associated tumors, oncogenic somatic alterations in MMR genes usually lead to impaired MMR function, characterized by dMMR, MSI-H, and the presence of mutational signature 6 [1,3,35]. However, breast cancer, which is not typically part of the Lynch syndrome tumor spectrum, raises uncertainty regarding whether somatic MMR alterations cause similar effects. Transcriptomic data were available for 1176 of the 3667 breast cancer samples (including 6 MLH1-altered, 5 MSH2-altered, 6 MSH6-altered, 5 PMS2-altered, and 1058 MMR-wt cases). We compared the mRNA expression levels of MMR genes between MMR-altered and MMR-wt breast cancers and found that MLH1 mRNA expression was significantly lower in MLH1-altered compared to MMR-wt breast cancers (Figure 2A). However, oncogenic somatic alterations in the other three MMR genes did not significantly affect the mRNA levels of their respective genes (Figure 2A). Furthermore, immunohistochemical staining of a breast cancer sample harboring an MLH1 mutation revealed complete loss of MLH1 and PMS2 protein expression in tumor cells (Figure 2B). MSI analysis based on NGS data showed that 27.0% of MMR-altered breast cancers exhibited MSI-H, which was significantly higher than the proportion observed in MMR-wt breast cancers (27.0% vs. 0.9%, *p* < 0.001, Chi Square test) (Figure 2C).

The above results indicate that somatic MMR alterations could lead to dMMR and MSI-H. Next, we examined the impact of somatic MMR alterations on the transcriptome profile of breast cancers. GSEA revealed that multiple DNA damage repair pathways were significantly upregulated in MMR-altered breast cancers, including the MMR, homologous recombination repair, and base excision repair pathway (Figure 2D,E). This suggests that somatic MMR alterations may contribute to genomic instability, triggering compensatory activation of various DNA repair pathways. We then analyzed the mutational signatures profiles of MMR-altered breast cancers (Figure 2F) and observed that mutational signature 3 and 6 were significantly enriched in MMR-altered compared to MMR-wt breast cancers (Figure 2G). These signatures are known to be associated with homologous recombination deficiency and MMR deficiency, respectively [35]. MMRDetect is a tool designed to identify MMR deficiency based on mutational signatures [30]. MMRDetect analysis using WES data revealed that 29.4% of MMR-altered breast cancers exhibited MMR deficiency (Appendix A), a proportion significantly higher than that observed in MMR-wt breast cancers (5/17 vs. 2/676, *p* < 0.001, Chi Square test) (Table 1), further supporting that somatic MMR alterations led to impaired function of DNA damage repair in cancer cells, particularly the MMR pathway.

Next, we investigated whether the type of mutation or biallelic inactivation in MMR-altered breast cancers differentially affects MMR function. Based on mutation types, the MMR alterations identified in this study (*N* = 44) were categorized as frameshift mutations (*N* = 6), nonsense mutations (*N* = 16), one deleterious missense mutation (*N* = 1), splice-site mutations (*N* = 6), structural variations causing gene fusions (*N* = 3), and copy number losses (*N* = 12) (Appendix A). We assessed the impact of these alteration types on MSI status and mutational signature 6 in breast cancer but found no significant differences (Appendix A). Among the 44 MMR-altered breast cancers, 12 cases with homozygous copy number loss in MMR genes and 9 cases with LOH of the wild-type allele were classified as MMR biallelic inactivation (Appendix A). Breast cancers with biallelic inactivation exhibited significantly elevated levels of mutational signature 6 (0.17 ± 0.25 vs. 0.04 ± 0.13, *p* = 0.01, Mann–Whitney U test) compared to those with monoallelic inactivation (Appendix A). 

### 3.3. Somatic Alterations in MMR Genes Enhance the Immunogenicity of Breast Cancer

MMR deficiencies could enhance the immunogenicity (e.g., TMB) and immunoreactivity (e.g., immune cell infiltration) in Lynch syndrome-associated tumors, thereby conferring their sensitivity to immunotherapy [3]. In this study, we investigated whether somatic MMR alterations affect the immune microenvironment of breast cancer. We observed that the TMB in MMR-altered breast cancers was significantly higher than in MMR-wt breast cancers (Median 3.7 vs. 0.5 mut/Mb, *p* < 0.001, Mann–Whitney U test) (Table 1) (Figure 3A). Notably, 29.5% of MMR-mutated breast cancers were classified as TMB-H (>10 mut/Mb), in stark contrast to only 2.0% in the MMR-wt cohort (29.5% vs. 2.0%, *p* < 0.001, Chi Square test). Subgroup analysis further revealed that MMR-altered breast cancers consistently exhibited a significantly higher TMB compared to MMR-wt breast cancers, regardless of MSI status (Figure 3A).

The number of neoantigens is considered a more representative indicator of tumor immunogenicity [36]. In alignment with the TMB results, we observed that the neoantigen load was significantly higher in the MMR-altered group compared to the MMR-wt group (median: 8.6 vs. 1.8 per Mb, *p* < 0.001, Mann–Whitney U test) (Table 1). Additionally, within the MMR-altered breast cancers, neither TMB nor neoantigen load showed any significant association with the mutation type or biallelic inactivation of MMR genes (Appendix A). These findings suggested that somatic MMR alterations significantly enhanced the immunogenicity of breast cancer. However, transcriptomic data revealed no significant differences in the mRNA expression levels of PD-1 and PD-L1 or in immune cell infiltration between MMR-altered and MMR-wt breast cancers (Figure 3B,C).

### 3.4. Comparison of the Genomic Landscape of MMR-Altered and MMR-wt Breast Cancers

We next compared the genomic landscape between MMR-altered and MMR-wt breast cancers. Compared to MMR-wt tumors, MMR-altered breast cancers exhibited a higher frequency of mutations in several key driver genes, including but not limited to TP53 (61.0% vs. 34.5%), KMT2C (26.8% vs. 6.0%), MLH1 (26.8% vs. 0.0%), PTEN (24.4% vs. 6.0%), and NF1 (22.0% vs. 4.1%) (Figure 4A,B). In terms of oncogenic copy number variations (CNVs), MMR-altered breast cancers displayed significantly increased frequencies of MYC amplification (38.5% vs. 20.3%), RAD21 amplification (35.9% vs. 16.9%), RECQL4 amplification (28.2% vs. 15.2%), PMS2 deletion (18.0% vs. 0.0%), FAT1 deletion (12.8% vs. 1.3%), and RAC1 deletion (12.8% vs. 0.04%) (Figure 4C,D). After adjusting for the molecular subtypes, the majority of the aforementioned somatic gene mutations remained significantly enriched in the MMR-altered group (Appendix A). Within the MMR-altered breast cancers, no significant differences in the genomic landscape were observed between cases with biallelic and monoallelic inactivation of MMR genes. These findings indicate that there are substantial genomic differences between MMR-altered and MMR-wt breast cancers.

### 3.5. Somatic Alterations in MMR Genes and Intratumoral Microbiota Characteristics in Breast Cancer

Recently, the intratumoral microbiota has been widely implicated in tumor development and progression [37,38]. Recent studies reported that MSI-H colorectal cancer showed distinct intratumoral microbiota [39,40], prompting us to explore whether MMR alterations or MSI status were associated with intratumoral microbiota features in breast cancer. We obtained microbial profiles from breast tumors utilizing transcriptomic data from the TCGA-BRCA dataset. Our results indicated that neither MMR alterations nor MSI status was associated with the alpha or beta diversity of the intratumoral microbiota in breast cancers (Figure 5A,B). We then examined the intratumoral microbial composition. After applying the most stringent filtering criteria, no significant differences were observed in the predominant microbial species between MMR-altered and MMR-wt breast cancers (Figure 5C). However, Pseudomonas was identified as the predominant species in MSI-H breast cancers, while it was not the most prevalent species in MSS breast cancers, suggesting a potential link between MSI-H status and microbiota composition in breast cancer.

## 4. Discussion

In this study, we conducted a comprehensive analysis of somatic MMR alterations in over 3000 breast cancer cases across six public cohorts. Our findings revealed that 1.2% of breast cancers harbored somatic MMR alterations, with TNBC exhibiting the highest mutation rate at 3.1%. Additionally, somatic MMR alterations were significantly associated with MSI-H and MMR-related mutational signatures. Importantly, these alterations significantly increased the TMB in breast cancer, irrespective of MSI-H status. Furthermore, MMR deficiencies in breast cancer may also influence the tumor mutational landscape and the composition of the intratumoral microbiome.

In Lynch syndrome-associated cancers, the predominant mechanisms of MMR gene inactivation are MMR germline variants and MLH1 promoter methylation [1]. This study found that the frequency of somatic MMR alterations in breast cancer was 1.2%, approximately eight times higher than previously reported rates of MMR germline variants (0.15%) [12]. Additionally, MLH1 promoter methylation was reported as extremely rare in breast cancer [41]. In this study, we only identified one case with MLH1 promoter methylation among 784 breast cancers from TCGA cohort. Notably, no somatic MMR alterations were detected in male breast cancer patients (0/31), which may be attributed to the relatively small sample size of male breast cancer cases. Future studies with larger cohorts of male breast cancer patients are needed to further investigate the frequency and significance of MMR alterations in this population. Collectively, these findings suggest that somatic MMR alterations represent the most prevalent form of MMR gene inactivation in breast cancer. Furthermore, we analyzed the frequency of somatic MMR alterations across various molecular subtypes and found that TNBC exhibited the highest mutation rate (3.1%). Currently, clinical trials investigating immunotherapy in breast cancer primarily focus on triple-negative subtypes, with discrepancy in efficacy among different stages, types of immunotherapy drugs, and PD-L1 status [42]; furthermore, the majority of these cases still do not benefit from immunotherapy [43]. PD-L1 expression serves as a predictive biomarker for immunotherapy response and guiding treatment decisions. Certainly, additional markers are needed for better patient selection and expanding the potential beneficiary population in PD-L1-negative cases. Thus, our study indicates that the relatively high frequency of MMR alterations in TNBC may serve as a prospective biomarker for immunotherapy.

Determining whether somatic MMR alterations lead to impaired function of the MMR system is crucial for assessing the oncogenic potential of these alterations [6]. In this study, we found that 27% of breast cancers harboring somatic MMR alterations exhibited MSI-H, which was comparable to the MSI-H rate reported for breast cancers with pathogenic germline MMR variants. Additionally, mutational signature analysis revealed a significant association between somatic MMR alterations and mutational signature 6 in breast cancer, which is well established as a specific signature of MMR deficiency [35]. Recent findings by Davies et al. demonstrated that mutational signature analysis applied to WGS is a more effective approach for identifying MMR-deficient tumors in breast cancer [44]. However, our study was limited to utilizing WES and panel-sequencing data. Despite this limitation, by integrating data from six high-quality studies, we achieved an MMR-deficient breast cancer detection rate of 1.2%, which is comparable to the 1.3% detection rate reported for mutational signature analysis using WGS data [44]. Within MMR-altered breast cancers, we observed that biallelic inactivation of MMR genes leads to more pronounced MMR deficiency compared to monoallelic inactivation. Although MMR deficiency (dMMR or MSI-H) has been established as a predictive biomarker for immunotherapy in advanced solid tumors, irrespective of tumor type [5], it is important to note that the number of breast cancer patients in these clinical trials was very limited. Our findings suggest that MMR-altered breast cancers, particularly those with biallelic inactivation, exhibit a more severely impaired MMR system. These results indicate that somatic MMR alterations, especially biallelic inactivation, may render these patients more likely to benefit from immunotherapy. However, further data collection is required in future clinical practice to validate the efficacy of immunotherapy in this subset of patients.

Elevated TMB and an activated immune microenvironment are known to confer sensitivity to immunotherapy in MMR-deficient colorectal and gastric cancers [3]. In this study, we observed that somatic MMR alterations significantly increased TMB in breast cancer, with up to 29% of MMR-altered breast cancers exhibiting TMB > 10 mut/Mb (TMB-H). Notably, this increase was independent of MSI-H status, suggesting that MMR alterations may elevate TMB regardless of their impact on MMR functionality. A higher TMB is expected to activate the immune microenvironment [3]. However, we found that the abundance of immune cells and the expression levels of PD-1/PD-L1 in MMR-altered breast cancer did not significantly differ from those in MMR-wt breast cancers. This discrepancy may be attributed to the limited sample size available for analyzing the immune microenvironment in this study.

This study reveals significant differences in the genomic landscape between MMR-altered and MMR-wt breast cancers, as TP53 mutation, PTEN mutations, NF1 mutations, MYC amplification, RAD21 amplification, and RECQL4 amplification were enriched in MMR-altered breast cancers. The higher proportion of TNBC among MMR-altered cases may be a crucial influencing factor, as TP53 mutations and MYC amplification are known to be enriched in TNBC [45]. After adjusting for molecular subtype, the majority of the aforementioned somatic gene mutations including PTEN mutations remained significantly enriched in the MMR-altered group. Interestingly, a similar observation was also reported in endometrioid cancers, in which PTEN loss/somatic mutation was closely associated with MMR deficiency [46,47]. Recent research has demonstrated significant differences in the intratumoral microbiota of MSI-H colorectal cancers compared to MSS counterparts, highlighting the interplay between genomic characteristics and intratumoral microbiota [39,40]. Similarly, our study found that MSI-H breast cancers exhibited distinct microbial compositions compared to MSS breast cancers, characterized by an enrichment of Pseudomonas, indicating that genomic features may influence the intratumoral microbiome in breast cancer. Future studies are needed to further validate this finding through experimental approaches.

There are several limitations in this study. Although we have compiled sequencing data from over 3000 breast cancer patients across multiple studies, the number of MMR-altered breast cancers was very small. Further independent breast cancer cohorts in the future are needed to validate our results. Secondly, some clinical information in this study remains incomplete, although we have tried our best to gather clinical data from public databases.

In conclusion, our study identifies 1.2% of breast cancers as harboring a somatic MMR alteration, with the highest mutation rate observed in the triple-negative subtype, at 3.1%. Somatic MMR alterations are significantly associated with MSI-H and MMR mutational signatures in breast cancers. These MMR alterations significantly increase the TMB of breast cancer, regardless of their impact on MSI-H status. Furthermore, MMR deficiencies may influence the mutational landscape and intratumoral microbiome. These findings enhance our understanding of the frequency, consequences, and clinical implications of somatic MMR alterations in breast cancer, suggesting that some patients with somatic MMR alterations may benefit from immunotherapy.

## Figures and Tables

**Figure 1 bioengineering-12-00426-f001:**
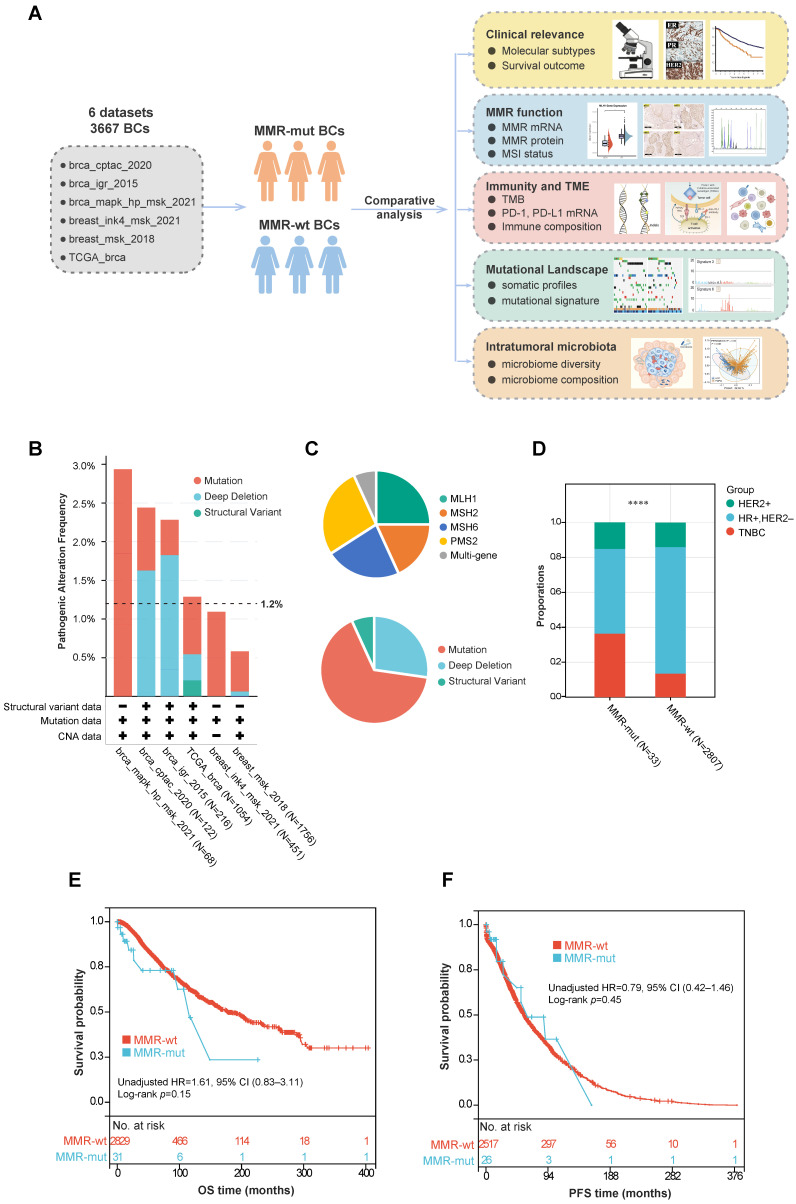
Study design and correlation between somatic MMR alterations and breast cancer phenotypes. (**A**) Flowchart of study design; (**B**) frequency of somatic MMR alterations across various public cohorts; (**C**) composition of various MMR gene and mutation types in breast cancer; (**D**) comparison of molecular subtypes between MMR-altered and MMR-wildtype breast cancers. The *p*-value was calculated using the Chi-Square test. **** *p* < 0.0001; (**E**,**F**) Kaplan–Meier survival analyses of MMR-altered versus MMR-wildtype breast cancer. The *p*-value was calculated using the Log rank test. Abbreviations: MMR, Mismatch Repair; CNV, Copy Number Variation; HR, Hormone Receptor; HER2, Human Epidermal Growth Factor Receptor 2; TNBC, Triple-Negative Breast Cancer; OS, Overall Survival; PFS, Progression-Free Survival.

**Figure 2 bioengineering-12-00426-f002:**
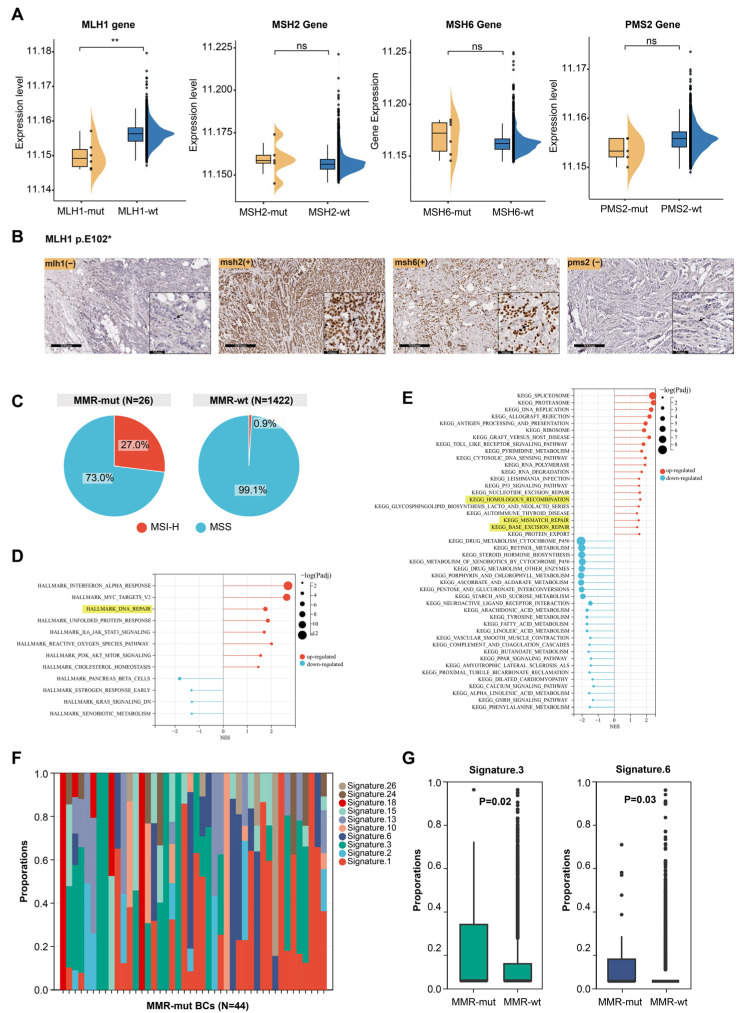
Effect of somatic MMR alteration on the function of MMR system. (**A**) Effect of somatic MMR alteration on mRNA expression levels of MMR genes. The *p*-value was calculated using the Mann–Whitney U test. ** *p* < 0.001. ns, not significant; (**B**) immunohistochemical staining of four MMR proteins in a breast cancer patient harboring MLH1 somatic mutation. Arrow pointed to breast cancer cells; (**C**) comparison of MSI status between MMR-altered and MMR-wildtype breast cancers; (**D**,**E**) GSEA of hallmark pathways and KEGG dataset between MMR-altered and MMR-wildtype breast cancers; (**F**) composition of mutational signatures in 44 MMR-altered breast cancers; (**G**) significantly different mutational signatures between MMR-altered and MMR-wildtype breast cancers. The *p*-value was calculated using the Mann–Whitney U test.

**Figure 3 bioengineering-12-00426-f003:**
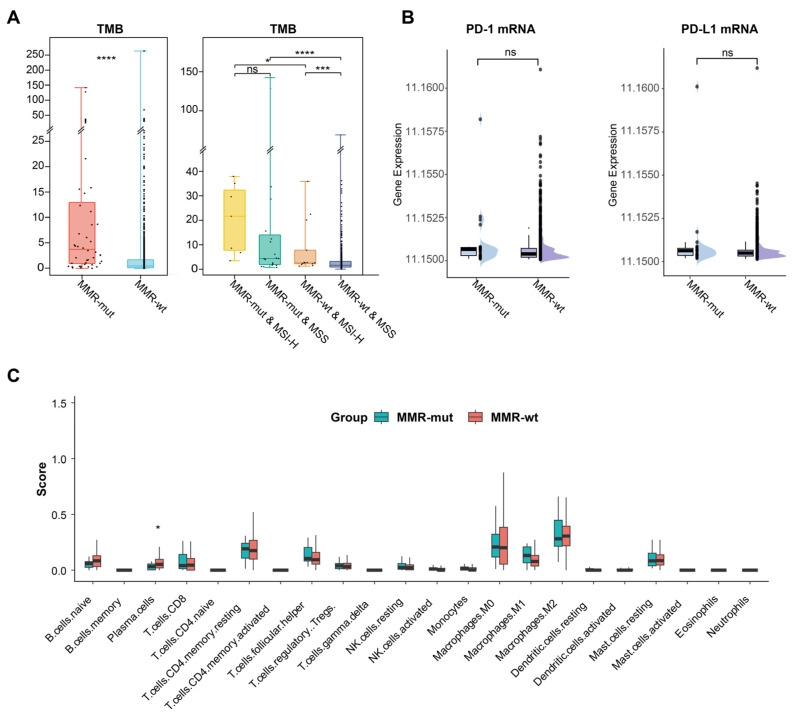
Impact of somatic MMR alterations on TMB and immune microenvironment in breast cancer. (**A**) Comparison of TMB between MMR-altered and MMR-wildtype breast cancers. The *p*-value was calculated using the Mann–Whitney U test. * *p* < 0.05. *** *p* < 0.001. **** *p* < 0.0001. ns, not significant; (**B**) comparison of PD-1 and PD-L1 mRNA level between MMR-altered and MMR-wildtype breast cancers. The *p*-value was calculated using the Mann–Whitney U test. ns, not significant; (**C**) comparison of various immune cell compositions in MMR-altered and MMR-wildtype breast cancers by CIBERSORT. The *p*-value was calculated using the Mann–Whitney U test. * *p* < 0.05.

**Figure 4 bioengineering-12-00426-f004:**
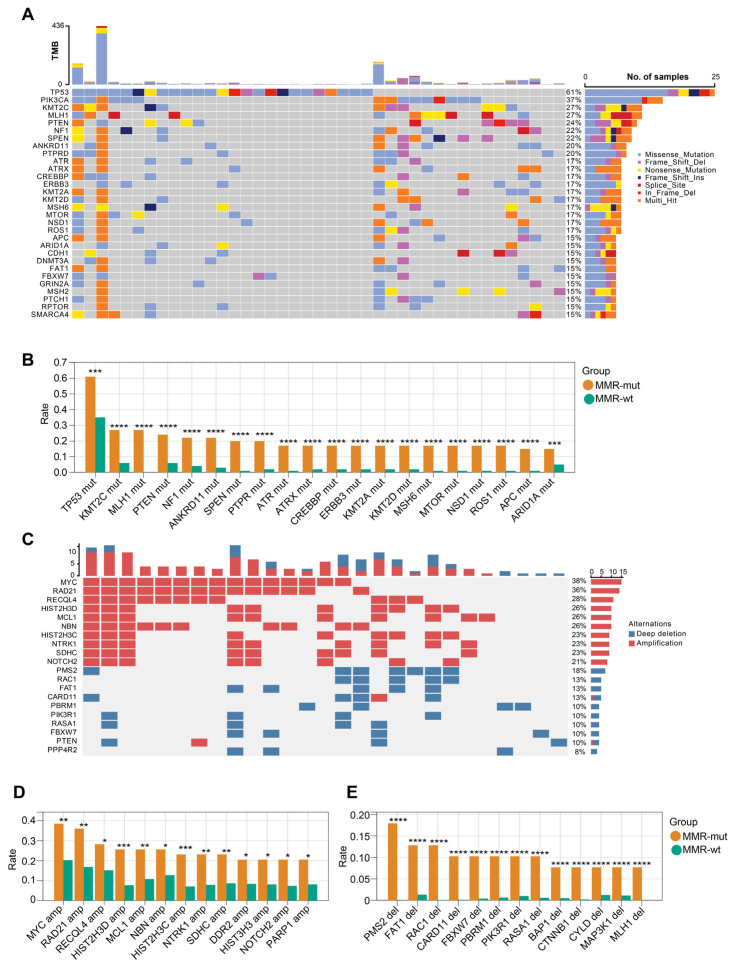
OncoPrint view of the genomic landscape of MMR-altered breast cancers. (**A**) Somatic mutation profiles of MMR-altered breast cancers. Genes are ordered by mutation rate; TMB (mut/Mb) is recorded on the top of the plot; (**B**) genes exhibiting significantly increased mutation rate in MMR-altered breast cancers compared to MMR-wildtype breast cancers. The *p*-value was calculated using the Chi Square test. *** *p* < 0.001. **** *p* < 0.0001; (**C**) profiles of somatic copy number changes in MMR-altered breast cancers; (**D**,**E**) genes exhibiting significantly increased rate of somatic copy number changes in MMR-altered breast cancers compared to MMR-wildtype breast cancers. The *p*-value was calculated using the Chi Square test. * *p* < 0.05. ** *p* < 0.01. *** *p* < 0.001. **** *p* < 0.0001.

**Figure 5 bioengineering-12-00426-f005:**
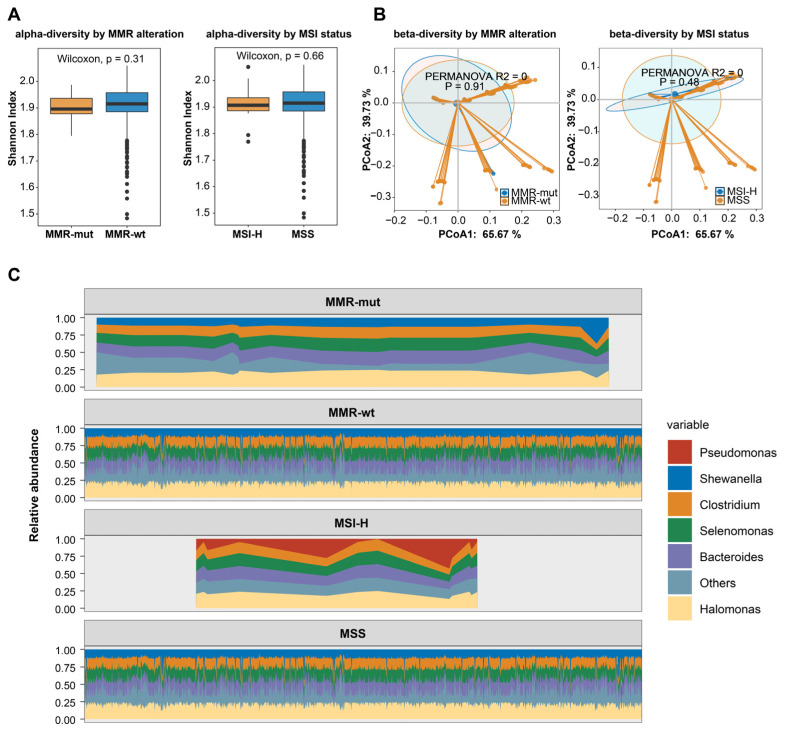
MMR deficiency and intratumoral microbiota characteristics in breast cancers. (**A**) The association of somatic MMR alterations and MSI status with the α-diversity of intratumoral microbiota in breast cancer. The *p*-value was calculated using the Mann–Whitney U test; (**B**) the association of somatic MMR alterations and MSI status with the β-diversity of intratumoral microbiota in breast cancer; (**C**) the association of somatic MMR alterations and MSI-H with the composition of intratumoral microbiota in breast cancer.

**Table 1 bioengineering-12-00426-t001:** Comparison of clinical characteristics between breast cancers with and without pathogenic somatic mutations in MMR genes.

	Total	MMR-Altered	MMR-wt	*p*-Value
No. of patients	3667	44	3623	
Age at diagnosis				0.01
Median (range)	54 (23–95)	57 (31–90)	54 (23–95)	
Mean ± SD	54.6 ± 13.0	60.3 ± 14.5	54.6 ± 13.0	
Ethnicity				0.7
White	1123 (77.2%)	19 (76.0%)	1104 (77.3%)	
African American	226 (15.5%)	5 (20.0%)	221 (15.4%)	
Asian	105 (7.2%)	1 (4.0%)	104 (7.2%)	
NA	2213	19	2194	
Gender				0.5
Female	3617 (99.2%)	42 (100.0%)	3575 (99.1%)	
Male	31 (0.8%)	0 (0.0%)	31 (0.9%)	
NA	19	2	17	
Histology				0.7
IDC	2200 (76.1%)	26 (86.7%)	2174 (76.0%)	
ILC	517 (17.9%)	2 (6.7%)	515 (18.0%)	
Medullary	5 (0.2%)	0 (0.0%)	5 (0.2%)	
Metaplastic	17 (0.6%)	0 (0.0%)	17 (0.6%)	
Mucinous	19 (0.7%)	0 (0.0%)	19 (0.7%)	
Mix	133 (4.6%)	2 (6.7%)	131 (4.6%)	
NA	776	14	762	
Molecular subtype				<0.001
HR + HER2-	2049 (72.1%)	16 (48.5%)	2033 (72.4%)	
HER2+	399 (14.0%)	5 (15.1%)	394 (14.0%)	
TNBC	392 (13.8%)	12 (36.4%)	380 (13.5%)	
NA	827	11	816	
TNM stage				0.9
Stage 0	1 (0.0%)	0 (0.0%)	1 (0.0%)	
Stage I	712 (24.2%)	7 (21.9%)	705 (24.3%)	
Stage II	1202 (40.9%)	15 (46.9%)	1187 (40.9%)	
Stage III	611 (20.8%)	7 (21.9%)	604 (20.8%)	
Stage IV	411 (14.0%)	3 (9.4%)	408 (14.0%)	
NA	730	12	718	
TMB (/Mb)				<0.001
Mean ± SD	1.8 ± 6.4	13.8 ± 28.9	1.6 ± 5.5	
Median (Range)	0.5 (0–264.6)	3.7 (0–142.4)	0.5 (0–264.6)	
TMB class				<0.001
TMB-H	84 (2.3%)	13 (29.5%)	71 (2.0%)	
TMB-L	3583 (97.7%)	31 (70.5%)	3552 (98.0)	
Neoantigen (/Mb)				<0.001
Mean ± SD	3.4 ± 7.8	27.0 ± 45.1	2.9 ± 3.8	
Median (Range)	1.9 (0–158.8)	8.6 (0–158.4)	1.9 (0–43.7)	
MSI status				<0.001
MSI-H	20 (1.4%)	7 (27.0%)	13 (0.9%)	
MSS	1428 (98.6%)	19 (73.0%)	1409 (99.1%)	
NA	2219	18	2201	
MMRDetect				<0.001
Pos	7 (1.0%)	5 (29.4%)	2 (0.3%)	
Neg	686 (99.0%)	12 (70.6%)	674 (99.7%)	
NA	2974	27	2947	
Mutation Sig6				0.003
Mean ± SD	0.04 ± 0.20	0.10 ± 0.20	0.04 ± 0.20	
Median (Range)	0 (0–1.0)	0 (0–0.7)	0 (0–1.0)	

Categorical variables were compared using the Chi Square test. Continuous variables were tested with the Mann–Whitney U test. Abbreviations: MMR, Mismatch Repair; SD, Standard Deviation; IDC, Invasive Ductal Carcinoma; ILC, Invasive Lobular Carcinoma; NA, Not Available or Not Applicable; HR, Hormone Receptor; HER2, Human Epidermal Growth Factor Receptor 2; TNBC, Triple-Negative Breast Cancer; TMB, Tumor Mutational Burden; MSI, Microsatellite Instability.

**Table 2 bioengineering-12-00426-t002:** Comparison of pathogenic somatic mutations in MMR genes among different molecular subtypes of breast cancer.

Molecular Subtypes	No. of Patients	MMR-Altered	Rate	*p*-Value
HR+, HER2−	2049	16	0.8%	<0.001
HER2+	399	5	1.3%	0.09
TNBC	392	12	3.1%	ref

The *p*-value was calculated using the Chi-Square test. Abbreviations: MMR, Mismatch Repair; HR, Hormone Receptor; HER2, Human Epidermal Growth Factor Receptor 2; TNBC, Triple-Negative Breast Cancer.

## Data Availability

All the genomic, transcriptome, and clinic data used in this study were obtained from public datasets as described in the Methods section. Any additional information required is available from the corresponding author upon reasonable request.

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
