# Peer review of "Comprehensive Analysis of Oncogenic Somatic Alterations of Mismatch Repair Gene in Breast Cancer Patients"

_bioengineering, 2025, doi:10.3390/bioengineering12040426_

Round 1

Reviewer 1 Report

Comments and Suggestions for Authors

Yan et al. presented results from a comprehensive survey of somatic MMR mutations in a large number (>3000) cancer cases. They found oncogenic MMR alterations in a portion of cases, especially triple negative cases. Although the percentage is relatively small (1.2% overall and 3.1% for triple negative), the finding is potentially useful as elevated TMB may sensitize cancers to immunotherapy.  This study could stimulate more extensive surveys to verify Yan et al.’s data and reveal more insights on the relationship between MMS mutations and breast cancers.

Major point:

“Introduction” lacks a summary of results at the end.

Minor points:

Line 15            change response to responsive

Line 23            change impair to impaired

Line 43            change arousing to aroused

Line 337          Table 36?

Comments on the Quality of English Language

There are quite a few grammatical mistakes in the text that should be corrected. 

Author Response

Comments 1: “Introduction” lacks a summary of results at the end.
Response 1: Agree. We have revised the final paragraph of the Introduction to provide a concise summary of the study’s results. (Page 2, Lines 57–62 in the revised manuscript).

Comments 2: Line 15 change response to responsive. Line 23 change impair to impaired. Line 43 change arousing to aroused.
Response 2: Thank you for pointing this out. We have made the revisions accordingly based on your suggestions. (Line 15, Line 23 and Line 43 in the revised manuscript).

Comments 3: Line 337 Table 36?
Response 3: Thank you for pointing this out. This error occurred during the editing process, where "36" should have been reference "[36]". We have corrected it accordingly and apologize for the oversight. (Page 12, Lines 341–342 in the revised manuscript).

Comments 4: There are quite a few grammatical mistakes in the text that should be corrected.
Response 4: Thank you for pointing this out. We have thoroughly revised the manuscript to correct all grammatical errors.

Reviewer 2 Report

Comments and Suggestions for Authors

The objective of the study was to provide new information to further characterize further the somatic alterations in MMR genes that age found in breast cancers (BC). The data for the analysis was essentially derived from online data acquisition sources listed in the study, with exception of MMR protein analysis in BC tissue from a single BC patient. Curated genes for panel sequencing included MMR genes MLH1, MSH2, MSH6, PMS2 and online analysis using WES, NGS, GSEAKEGG, and CIBERSORT.

The Research and Ethics Committee of Peking University Cancer Hospital approved procedures of the study, and written consent from the single BC was reported.

Data records from over 3,000 breast cancer cases across six public cohorts were included in the analysis. Several male BC data (31 cases) were included, none of which had MMR alternations. This gender-related result deserves a comment in discussion.

A complete list of the types infiltrating immune cells should be listed (line 99).

Methods are generally well described. Adequate details are provided. Identifying vendors and reagents (e.g., antibody clones)  are well apricated and could be valuable to subsequent investigations of others.

Statistical/data tests included Chi Square test  Fisher’s exact  test, Kaplan-Meier method,  Mann-Whitney U test, Log  rank test, and t-test. The test type that determined P-values listed in the experiment results/figure legends should be indicated.

Figure 2

Quantified mRNA levels for  MLH1, MSH2, MSH6, and PMS2  are shown in Fig. 2A.

Are data points repeat outcomes of the same sample, or outcomes from samples of different patients?

Plotting gene transcript levels for MLH1-wt, MSH2-wt, MSH6-wt, and PMS2-wt would be more useful that showing levels of MMR-wt.

Although gene transcripts for MLH1, MSH2, MSH6, and PMS2  are evident (Fig. 2A), antibodies did not detect MLH1 and PMS2 protein. This result (mRNA detected, protein product absent) highlights issues with methodology and experiment design interpretation

Shown is tissue from one patient (n=1) examined for MLH1, MSH2, MSH6, and PMS2  protein.

The result could be interesting (positive for MSH2, MSH6, negative for MLH1 and PMS2), however the n=1 sample is unreliable. Additionally, this unexpected result is not mentioned or analyzed further. The result should be included in the discussion section. Additional BC tumor samples of different banked sources of BC should must be included in the data image set with analysis and statistical evaluation.

Immunological staining for MSH2 and MSH6 is not clearly nuclear vs nuclear/cytoplasmic/extracellular. Increased magnification of the immunohistochemical images should be shown, including indicators of stained and non-stained nuclei. Scale bar should be included.

An unrelated negative control antibody should be included in the image data set. Likewise, a positive control tissue should be included in order to demonstrate confidence in the MSH2 and PMS2  antibodies.

Authors indicate some of limitations of their study, including a relatively modest data set derived from a small number of MMR altered BC.

The conclusion that somatic MMR alternations that inactivate MMR genes are prevalent in BC is consistent with the data.

Author Response

Comments 1: Data records from over 3,000 breast cancer cases across six public cohorts were included in the analysis. Several male BC data (31 cases) were included, none of which had MMR alternations. This gender-related result deserves a comment in discussion.
Response 1: We sincerely appreciate your valuable comments. We have incorporated a discussion on the absence of MMR alterations in male breast cancer patients in the Discussion section. We believe that the lack of detected somatic MMR alterations in male breast cancer patients (0/31) may be attributed to the relatively small sample size of male breast cancers in this study. Future studies with larger cohorts of male breast cancer patients are needed to further investigate the frequency and significance of MMR alterations in this population.  (Page 16, Lines 424–428 in the revised manuscript).

Comments 2: A complete list of the types infiltrating immune cells should be listed (line 99).
Response 2: Thank you for pointing this out. We have supplemented a complete list of the types of infiltrating immune cells in the revised manuscript. (Page 3, Lines 101–107 in the revised manuscript).

Comments 3: Statistical/data tests included Chi Square test Fisher’s exact test, Kaplan-Meier method, Mann-Whitney U test, Log rank test, and t-test. The test type that determined P-values listed in the experiment results/figure legends should be indicated.
Response 3: Thank you for pointing this out. We have incorporated the corresponding statistical test method in the sections describing p-values in the results and figure legends.

Comments 4: Figure 2 Quantified mRNA levels for MLH1, MSH2, MSH6, and PMS2 are shown in Fig. 2A. Are data points repeat outcomes of the same sample, or outcomes from samples of different patients?
Response 4: The data points represent outcomes from different patient samples, with each point corresponding to an individual patient.

Comments 5: Plotting gene transcript levels for MLH1-wt, MSH2-wt, MSH6-wt, and PMS2-wt would be more useful that showing levels of MMR-wt.
Response 5: Agree. We have replotted the gene transcript levels for MLH1-wt, MSH2-wt, MSH6-wt, and PMS2-wt based on your suggestion. (Figure 2A in the revised manuscript)

Comments 6: Although gene transcripts for MLH1, MSH2, MSH6, and PMS2 are evident (Fig. 2A), antibodies did not detect MLH1 and PMS2 protein. This result (mRNA detected, protein product absent) highlights issues with methodology and experiment design interpretation.
Response 6: Thank you for pointing this out. Figure 2A is based on transcriptomic data from the TCGA cohort, while Figure 2B represents a single breast cancer sample from our research center. We acknowledge that a single case lacks representativeness; however, this example only aims to illustrate that MMR-altered breast cancer can exhibit the loss of the corresponding protein.

Comments 7: Shown is tissue from one patient (n=1) examined for MLH1, MSH2, MSH6, and PMS2 protein. The result could be interesting (positive for MSH2, MSH6, negative for MLH1 and PMS2), however the n=1 sample is unreliable. Additionally, this unexpected result is not mentioned or analyzed further. The result should be included in the discussion section. Additional BC tumor samples of different banked sources of BC should must be included in the data image set with analysis and statistical evaluation.
Response 7: Thank you for your comments. Due to the low frequency of MMR alterations in breast cancer and the absence of MMR genes from most clinical gene testing panels for breast cancer, only one breast cancer patient with an MMR mutation was identified and sequenced at our treatment center. The data presented in Figure 2A were obtained from the TCGA public database; therefore, MMR immunohistochemical analysis could not be performed on these tumor samples.

Comments 8: Immunological staining for MSH2 and MSH6 is not clearly nuclear vs nuclear/cytoplasmic/extracellular. Increased magnification of the immunohistochemical images should be shown, including indicators of stained and non-stained nuclei. Scale bar should be included. 
Response 8: Thank you for your suggestion. We have addressed this in the revised version. (Figure 2B in the revised manuscript)

Comments 9: An unrelated negative control antibody should be included in the image data set. Likewise, a positive control tissue should be included in order to demonstrate confidence in the MSH2 and PMS2 antibodies. 
Response 9: Thank you for your suggestion. We have addressed this in the revised manuscript. (Figure S1 in the revised manuscript)

Reviewer 3 Report

Comments and Suggestions for Authors

The manuscript is well drafted, the methods are reasonable, and results are good and discussion is well organized.

Comments on the Quality of English Language

ok need some spelling errors and minor corrections needed. For instance, in the table, Ethnic has been spelled as Ethic. 

Author Response

Comments 1: ok need some spelling errors and minor corrections needed. For instance, in the table, Ethnic has been spelled as Ethic.  
Response 1: Thank you for pointing this out. We have carefully reviewed the entire manuscript for grammar and spelling, making every effort to correct any spelling errors.

Round 2

Reviewer 2 Report

Comments and Suggestions for Authors

Comments/suggestions have been satisfactorily addressed.